# Geographical Disparities and Patients’ Mobility: A Plea for Regionalization of Pancreatic Surgery in Italy

**DOI:** 10.3390/cancers15092429

**Published:** 2023-04-24

**Authors:** Gianpaolo Balzano, Giovanni Guarneri, Nicolò Pecorelli, Stefano Partelli, Stefano Crippa, Augusto Vico, Massimo Falconi, Giovanni Baglio

**Affiliations:** 1Pancreatic Surgery Unit, Pancreas Translational and Clinical Research Center, IRCCS San Raffaele, 20132 Milan, Italy; guarneri.giovanni@hsr.it (G.G.); falconi.massimo@hsr.it (M.F.); 2Pancreatic Surgery Unit, Pancreas Translational and Clinical Research Center, San Raffaele Scientific Institute, Vita-Salute San Raffaele University, 20132 Milan, Italy; 3Codice Viola, Patient Advocacy Organization; 4Head of the Research and International Relations Unit, Italian National Agency for Regional Healthcare Services, 00187 Rome, Italy

**Keywords:** pancreatic surgery, regionalization, patients’ mobility, hospital volume

## Abstract

**Simple Summary:**

This study highlights significant issues with access to pancreatic surgery in Italy. The distribution of adequate facilities for pancreatic surgery is inhomogeneous throughout the country, with provision decreasing from north to south. This study found that patients from Southern and Central Italy had significantly higher rates of mobility to Northern Italy for pancreatic resections compared to patients in the north. This highlights the unequal access to pancreatic surgery that exists in Italy. This study also found that mortality rates for non-migrating patients receiving surgery in Southern and Central Italy were significantly higher than for migrating patients. This indicates the need to implement adequate facilities in the southern and central regions, to reduce mortality and ensure equal access to care for all patients.

**Abstract:**

Patients requiring complex treatments, such as pancreatic surgery, may need to travel long distances and spend extended periods of time away from home, particularly when healthcare provision is geographically dispersed. This raises concerns about equal access to care. Italy is administratively divided into 21 separate territories, which are heterogeneous in terms of healthcare quality, with provision generally decreasing from north to south. This study aimed to evaluate the distribution of adequate facilities for pancreatic surgery, quantify the phenomenon of long-distance mobility for pancreatic resections, and measure its effect on operative mortality. Data refer to patients undergoing pancreatic resections (in the period 2014–2016). The assessment of adequate facilities for pancreatic surgery, based on volume and outcome, confirmed the inhomogeneous distribution throughout Italy. The migration rate from Southern and Central Italy was 40.3% and 14.6%, respectively, with patients mainly directed towards high-volume centers in Northern Italy. Adjusted mortality for non-migrating patients receiving surgery in Southern and Central Italy was significantly higher than that for migrating patients. Adjusted mortality varied greatly among regions, ranging from 3.2% to 16.4%. Overall, this study highlights the urgent need to address the geographical disparities in pancreatic surgery provision in Italy and ensure equal access to care for all patients.

## 1. Introduction

Operative mortality after pancreatic surgery varies significantly between centers, and is primarily dependent on hospital and/or surgeon volume [1,2] as well as the ability to reduce failure-to-rescue [3]. The first studies on the relationship between volume and outcome in pancreatic surgery were conducted at the end of the last century [4]. In Italy, the first nationwide analysis was published in 2008 [5]. Since then, both patient associations and high-volume centers have encouraged patients to seek out appropriate hospitals when pancreatic surgery is required. As a result, even in the absence of a national centralization policy, an increasing number of Italian patients requiring pancreatic surgery are turning to specialized centers. Between 2014 and 2016, 6153 pancreatic resections (48.4%) were performed in the 20 highest-volume facilities (5% of Italian hospitals performing pancreatic resections), while the remaining operations (6509) were performed in 375 hospitals [6].

The Italian National Health Service is a tax-funded, universal health system that is highly decentralized through region-based delivery systems. There are 21 health systems (19 regions and two autonomous provinces) that are independently responsible for delivering healthcare to residents. Italian citizens have the freedom to choose their providers, and they can move for medical care from one region to another. The socio-economic and healthcare conditions in Italy are inhomogeneous, generally worsening from north to south [7,8,9]. The regions in each geographical area (Northern, Central, and Southern Italy) share similar socio-economic and health indicators. The perception of inadequate healthcare provision by residents in southern and, to a lesser extent, central regions has resulted in a significant phenomenon of patient mobility for complex care, mainly towards northern regions [10]. This is particularly true for pancreatic surgery due to a lack of high-volume facilities outside of Northern Italy. The disparity in healthcare provision across regions raises concerns about equal access to care [11], particularly for those with limited financial resources who cannot afford travel expenses [12,13], or for patients unfit to travel due to more severe health conditions.

The aim of this study was to investigate the phenomenon of mobility for pancreatic resections in Italy, measure the effects of patients’ mobility on outcomes, and evaluate the adequacy of pancreatic surgery provision in each Italian region.

## 2. Materials and Methods

### 2.1. Study Design

This is a retrospective observational study based on administrative data. The dataset on pancreatic resections was obtained from the Directorate General of Health Planning of the Ministry of Health, whose database comprises all admissions in all Italian hospitals. Data refer to a three-year period (2014–2016) and were made anonymous by the Directorate in compliance with the rules on the protection of patients’ privacy. The following operations were included in the analysis: pancreaticoduodenectomy (ICD-9 codes: 52.7, 52.51), distal pancreatectomy (52.52), total pancreatectomy (52.6), and other resections (52.22, 52.53, 52.59). To reduce bias due to coding inaccuracy, operations with diagnosis unrelated to pancreatic surgery were excluded; urgent operations and operations for acute pancreatitis were also excluded because of possible non-standardized procedures. The dataset included the patients’ municipality of residence and the hospital where patients received surgery. The study was conducted and reported according to the RECORD statement [14]. The RECORD checklist is available in the Appendix A.

### 2.2. Study Objectives

The primary objectives were:To quantify the phenomenon of patients’ mobility to receive pancreatic surgery in Italy, comparing the characteristics of migrating and non-migrating patients and evaluating the travel rate and the attraction rate of each region;To analyze the effect of mobility on patients’ outcome;To investigate the presence of geographical disparities in the delivery of pancreatic surgery, measuring the operative mortality in each region and the availability of adequate facilities.

### 2.3. Methods and Definitions

The Italian territory is conventionally divided in three geographical areas: Northern, Central, and Southern Italy. Territorial division into regions and areas is reported in Figure 1. The two autonomous provinces (Trento and Bolzano) were gathered into the Trentino-Alto Adige region. The surgical outcomes of resections performed in the two smallest regions (Valle d’Aosta and Molise) are not reported because fewer than 10 resections/year were performed.

Operative mortality was defined as in-hospital mortality (the only data available in the archive). For each individual hospital performing ≥10 resections/year, both observed mortality and adjusted mortality rates (risk-standardized mortality rate—RSMR) were calculated. RSMR has been suggested as the best method for measuring the surgical quality of different hospitals in the same health system [15] and has already been applied to Italian data [6]. It considers covariates significantly affecting the risk of death (see below). Geographical disparities were assessed through the adjusted mortality rate (RSMR) of each region and geographical area and through the evaluation of the performance of each single hospital performing pancreatic resections in the different regions and geographic areas (Northern, Central, and Southern Italy). The number of facilities per million inhabitants offering a minimum volume of 10 resections/year and with an RSMR ≤ 5% and between 5 and 10% was calculated for each region and geographic area. The threshold of 10 resections/year as a minimum volume was selected because it was the lowest among the volume requirements accepted by centralization policies in Europe or North America [16].

Patients were defined as non-migrating if the region of residence was the same as where they received surgery. When patients were operated on in a region other than the one in which they resided, we distinguished between short- and long-distance mobility. Short-distance mobility was considered when the facility providing surgery was in another region, but it was closer than one hour’s travel time from the residence. In these cases, mobility towards a different region was considered to be driven by proximity reasons and patients were not considered as migrating. Only if the hospital was both in another region and farther than one hour from the residence were patients considered as migrating.

For each region and for each geographical area, the escape rate and the attraction rate were calculated: the escape rate was defined as the ratio between patients who received surgery outside the region/area of residence and overall patients who underwent resection residing in the same region/area. The attraction rate was defined as the ratio between operated patients residing in another region/area and overall patients operated on in the region/area.

### 2.4. Statistical Analysis

Analysis was performed using STATA^®^ version 14.0 (Stata Corp, College Station, TX, USA). The Pearson χ2 test was used for the comparison of categorical variables and ANOVA for continuous variables. Operative mortality was calculated for each region and each geographical area. The mortality of migrating and non-migrating patients was compared through uni- and multivariate analysis. The following factors were analyzed for a possible relation to mortality: hospital volume, comorbidities, gender, age, type of resection, main diagnosis, vascular resections, multivisceral resections, NHS agreement (private patients or not), and patient mobility.

Multivariate analysis for factors influencing in-hospital mortality was conducted using logistic regression. The model was obtained through a stepwise backward elimination entering all the variables analyzed in univariate (Wald method with *p* > 0.010 for removal from the model).

To classify hospitals according to pancreatic surgery volume, facilities were listed based on the ascending number of resections, and cut-off points were created to obtain classes with a similar number of patients, as already described [6]. Comorbidities were derived from ICD-9 codes and classified according to the Charlson–Deyo score [17], modified as suggested in 2005 [18]. Codes related to patients’ cancer were not considered to define the score. To account for patient characteristics and type of operation, RSMR was calculated, according to a previously described model [6] summarized here:

1. For each patient, an individual mortality risk was estimated using a multivariable logistic regression model including all independent variables, except for hospital volume and patient mobility; because of the large sample size, no preselection of variables in univariate analysis was defined. 2. The predicted probabilities of mortality for each patient were summed within each individual hospital, region, or area to obtain the expected number of deaths [19]. 3. The observed to expected mortality ratio (standardized mortality ratio—SMR) was calculated. 4. Finally, the risk-standardized mortality rate (RSMR) was obtained by multiplying the SMR by the overall observed mortality rate in Italy, to allow the comparison of the performance with the national average [20]. For the calculation of RSMR, when considering individual hospitals, only facilities with at least ten resections over 3 years were included, to reduce the bias due to the excessive RSMR variations due to hospitals with a very low number of resections. The adjusted operative mortality (RSMR) was used to assess the adequacy of facilities performing pancreatic surgery in each region. The estimation of patients’ travel time to reach the hospital by car was calculated through a distance matrix provided by the Italian national statistical institute (ISTAT) [21]. ISTAT provides the origin–destination matrices of the distances in meters and travel times in minutes between Italian municipalities through a geographic information system (GIS) software, using a commercial road graph, based on the average travel speeds of each road arc constituting the road graph.

All statistical tests were two-sided, and a value of *p* < 0.050 was considered statistically significant.

## 3. Results

The database included 12,844 pancreatic resections performed in 395 hospitals in Italy from 2014 to 2016. Of these, 182 patients were excluded due to possible coding errors, so the study group consisted of 12,662 patients. Hospitals were divided in five volume categories with increasing number of resections/year, as already described [6]: very low volume ≤ 10, low volume > 10–25, median volume > 25–60, high volume > 60–167, very high volume > 167. Most operations were performed in Northern Italy (63%), where about 45% of the Italian population resides. The rate of pancreatic resections per inhabitants progressively decreased from Northern to Central to Southern Italy (*p* < 0.001) (Table 1).

### 3.1. Patients’ Mobility

Overall, inter-regional mobility was observed in 2877 patients (22.4%). Of these, 274 patients (2.1%) traveled to proximity hospitals in neighboring regions (<one hour travel time, short-distance mobility) and were grouped with non-migrating patients. Long-distance mobility occurred in 2603 patients (20.3%); these patients were defined as migrating.

The overall escape rate from Southern Italy was 40.3% (1326 patients), mainly heading to Northern Italy (78.6% of them, 1085 patients). All southern regions showed escape rates higher than 30%, with peaks in Calabria (76.7%) and Abruzzo (52.6%) (excluding the small region of Molise). Mobility from Central to Northern Italy occurred at a lesser extent (overall escape rate 14.6%, 388 patients), whereas the escape rate from Northern Italy was negligible (1.1%, 75 patients). In Central Italy, the highest regional escape rate was observed in Marche (38.5%); in Northern Italy, it was observed in Liguria (41.5%) (excluding the small region of Valle d’Aosta). The regions with the highest attraction rate were Veneto (54.4%) and Lombardy (26%) (Table 1).

Table 2 compares the characteristics of migrating and non-migrating patients. Most migrating patients headed to very high-volume hospitals (59%), whereas only 12.9% of non-migrating patients received operations in such hospitals (*p* < 0.001). Further, migrating patients were younger, had a lower comorbidity index, less frequently underwent vascular or multivisceral resections, and more frequently had private reimbursement and diagnoses other than pancreatic cancer or chronic pancreatitis.

### 3.2. Mortality Analysis

The overall operative mortality on a national basis was 6.2%. The mortality rate was inversely related to hospital volume; this finding was previously described in detail [6]. Uni- and multivariate analyses of factors associated with mortality are shown in Table 3. Factors associated with an increased risk in univariate analysis were age > 70, male gender, Charlson comorbidity score ≥ 2, lower-volume hospital, major pancreatic resection, and associated vascular resection; as regards factors related to mobility, univariate analysis showed that residing in Southern and Central Italy, undergoing surgery in Southern and Central Italy, and not migrating for surgery were associated with an increased risk of operative mortality. In multivariate analysis, residing in Central or Southern Italy did not confirm an independent effect on mortality, as was also the case for vascular resections.

The expected mortality rates, estimated from patients’ and operations’ characteristics, were similar in different regions and areas, whereas the actual mortality rates were highly variable, with a general trend towards an odds increase from Northern to Central to Southern Italy (Figure 2). In Northern Italy, expected and observed mortality were 6.2% and 5.0%, in Central Italy, they were 6.2% and 6.9% (*p* < 0.05), and in the south they were 6.4% and 10.1%, (*p* < 0.001), respectively. As regards the mortality rates of single regions, RSMRs were quite heterogeneous, ranging from 3.2% in Veneto to 16.4% in Campania (Table 4). Liguria was the only northern region with a regional mortality rate >10% (RSMR 12.2%).

### 3.3. Distribution of Facilities in Geographical Areas and Regions

The number of facilities offering pancreatic surgery is reported in Table 5. Among them, we identified those offering acceptable performance (based on volume and mortality) and their distribution in different regions and areas. Ninety-two hospitals with a minimum volume of 10 resections/year were found, of which 75 had adjusted mortality rates <10% (sub-categorized into RSMR ≤ 5%, and between 5 and 10). Most of them were located in Northern Italy (46 hospitals, 63.9%), while 15 were located in Central and 14 in Southern Italy. When restricting facilities to RSMR ≤ 5%, 44 providers were identified, 32 of them located in the north (72.8%), 5 in the center and 7 in the south. The analysis of the number of facilities per million inhabitants confirmed the inhomogeneous distribution throughout Italy: the rate of adequate hospitals (annual volume ≥ 10 and RSMR ≤ 5%) per million inhabitants was 1.15 in Northern-, 0.42 in Central, and 0.34 in Southern Italy. Several regions, mainly located in the center–south, with the exception of Liguria, located in the north, had no availability of hospitals with optimal performance.

### 3.4. Effect of Mobility on Outcome

Operative mortality in non-migrating patients was higher than migrating patients (RSMR: 7.3% vs. 1.6%, respectively). Table 4 shows the mortality rates per region and geographical areas, according to the patients’ migrating status. Among patients residing in Southern Italy, non-migrating patients had 10.3% adjusted mortality, significantly higher than that of migrating patients (1.7%) (*p* < 0.001). To a smaller degree, the same effect was observed for patients migrating from Central to Northern Italy: the RSMR of non-migrating patients was 7.9%, whereas it was 1.7% for migrating patients (*p* < 0.01). Figure 3 summarizes the impact of patients’ mobility from Southern and Central Italy. In all the areas and in almost all the regions, the mortality of residing patients was significantly higher than that of patients who migrated from other regions (Table 4).

## 4. Discussion

Pancreatic surgery is among the most complex abdominal operations, and previous studies documented a clear correlation between the volume of surgery and the outcomes achieved [22,23]. In many countries, efforts to centralize these operations are ongoing [16], but despite a few rare exceptions [24], the results have been unsatisfactory. In Italy, no national policy for pancreatic surgery centralization has been defined, although criteria for hospital selection have been published previously [25]. Italy has a population of approximately 60 million people and exhibits extensive geographic, socio-economic, and healthcare disparities.

The study findings revealed that the current surgical offerings are unevenly distributed throughout the country, exposing the population to unequal access to high-quality care with severe clinical and economic consequences. Suitable facilities for pancreatic surgery throughout the country were identified, considering both the minimum volume and a low mortality rate. In fact, the minimum volume alone is not sufficient to ensure low operative mortality rates; a recent analysis in Italy showed that about half of the facilities with a minimum volume of 10 pancreatic resections per year had a mortality rate greater than 5%, and approximately 20% of them exceeded 10% [6]. For the present analysis, we selected adequate hospitals offering at least 10 resections per year with a low mortality rate (RSMR less than 5% or 10%). Out of 44 Italian hospitals meeting both the volume and mortality (<5%) requirements, most were located in Northern Italy (32 out of 44, 72%), and there was a lack of adequate facilities in several regions, mainly in Central and Southern Italy.

Thus, approximately 40% of patients residing in Southern Italy and 15% in Central Italy moved to northern regions in search of proper care, travelling several hundred kilometers and incurring significant private expenses to receive treatment away from home. More than one third of patients from every region of Southern Italy migrated for treatment, with up to 76% of patients migrating from Calabria. Outside Southern Italy, high escape rates were observed in Marche and Liguria. The highest attraction rates were observed in Veneto (54.4%) and Lombardy (26.6%).

Patients migrate for surgery because they are aware of better outcomes in specialized centers, and this information is readily available on the internet. In the context of pancreatic surgery, there is a fourfold reduction in the risk of operative mortality when surgery is provided in the highest-volume hospitals compared to the lowest-volume hospitals (20). Migrating for surgery reduced the risk of mortality. The adjusted mortality rate was 1.6% for migrating patients and 7.3% for non-migrating patients, and this finding was consistent across all regions and areas (with the exception of Umbria, a small region in Central Italy). The highest mortality rate for non-migrating patients was found in Campania (RSMR 17.2%), whereas in the same region, the RSMR was 1.6% for migrating patients. On a national basis, the majority of migrating patients (76.8%) headed towards high-volume centers with low mortality rates, whereas only a minority of resident patients (31.4%) were treated in such hospitals. Conversely, 44.7% of non-migrating patients were operated on in low- or very low-volume facilities, versus 12% of migrating patients. This is the effect of the lack of a centralization policy defined at an institutional level. We investigated the differences in patient characteristics between those who migrated and those who did not. As expected, migrating patients tended to be younger and had fewer comorbidities than non-migrating patients. This suggests that access to care outside one’s region of residence is dependent on individual characteristics, with sicker patients more likely to remain in their region despite the lack of adequate facilities. However, worse clinical characteristics only partially account for the higher operative mortality observed in non-migrating patients. The expected mortality rates calculated on specific patient characteristics were similar across different regions, but observed mortality showed great variability across regions and geographical areas. Likewise, in multivariate analysis, the odds ratio for migrating patients was 0.698 (95% confidence intervals: 0.54–0.90, *p* = 0.006).

These findings raise concerns about equity in access to pancreatic surgery in Italy. Patients seeking high-quality care may travel long distances [26], but this can have financial implications for patients and caregivers. Previous studies have shown that elderly patients and those in low socio-economic categories are less likely to travel for treatment [27]. Patient migration can create inequalities due to worse outcomes expected in nearby low-volume centers.

Another problem stemming from patient mobility from Southern/Central to Northern Italy is the transfer of financial resources from poorer to richer regions. Although the National Health Service guarantees patient mobility, each region has to pay for the treatment provided to residents by facilities located in other regions. This transfer of money from the poorest to the richest regions reduces available funds to improve health services in regions where services are lacking.

Our study also found a north–south gradient in the annual number of pancreatic resections per million inhabitants, with higher rates in Northern Italy. This was partially expected due to a higher incidence of pancreatic cancer in the north [28]. However, other factors such as a lower resection rate of pancreatic and periampullary malignancies due to the lack of high-volume centers, or a lower detection rate of pre-malignant neoplasms for less efficient healthcare services, could also play a role.

Our study has several limitations. First, the data date back to 2014–2016, and individual hospital outcomes may have changed over the years. Second, we used administrative data to derive clinical outcomes, which may be subject to errors or heterogeneity in coding practices. However, high-cost procedures, such as major operations, are usually coded correctly [29]. Finally, we used in-hospital mortality as the only available data in the archive, although 90-day mortality would better reflect the actual operative mortality in pancreatic surgery [30].

## 5. Conclusions

In conclusion, the current provision of pancreatic surgery throughout Italy is inadequate due to a lack of regulation. The shortage of adequate centers, particularly in southern regions, creates a phenomenon of healthcare migration, which exposes patients who cannot travel to a significantly increased risk of mortality. This phenomenon is a form of discrimination against groups with lower socio-economic status and patients with greater frailty. The present article highlights the healthcare planning issues mainly affecting southern regions, which is both an ethical and political problem. In Italy, healthcare services are organized independently by each region within its own borders, making it necessary to implement a centralization process at a regional level. Some regions have already started the process of identifying appropriate centers for providing pancreatic surgery, with Lombardy being the region with the most advanced project [31], also as a result of a recent analysis of the outcomes of pancreatic surgery in Lombardy [32]. To help healthcare institutions centralize pancreatic surgery, a new method has been proposed, based on criteria such as minimum surgical volume, low operative mortality, adequate territorial distribution, and impact on waiting lists [32].

Given the dire consequences of inadequate pancreatic surgery, the centralization of surgery is the most pressing issue, but this is just the first step towards improving the treatment of pancreatic diseases, which requires a multidisciplinary team with high-level specific skills.

## Figures and Tables

**Figure 1 cancers-15-02429-f001:**
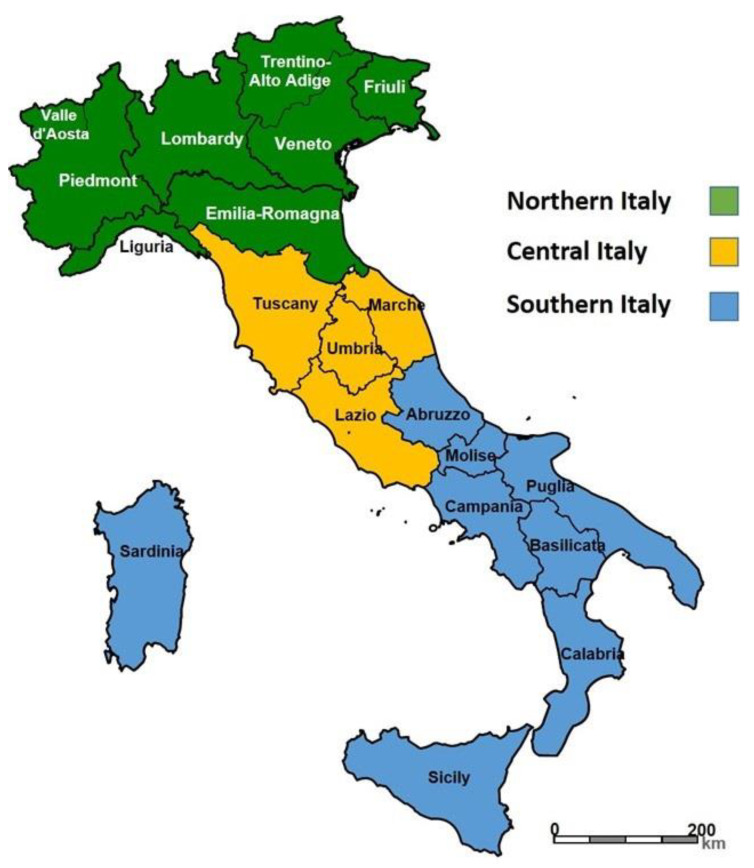
Italy territorial division into regions and areas.

**Figure 2 cancers-15-02429-f002:**
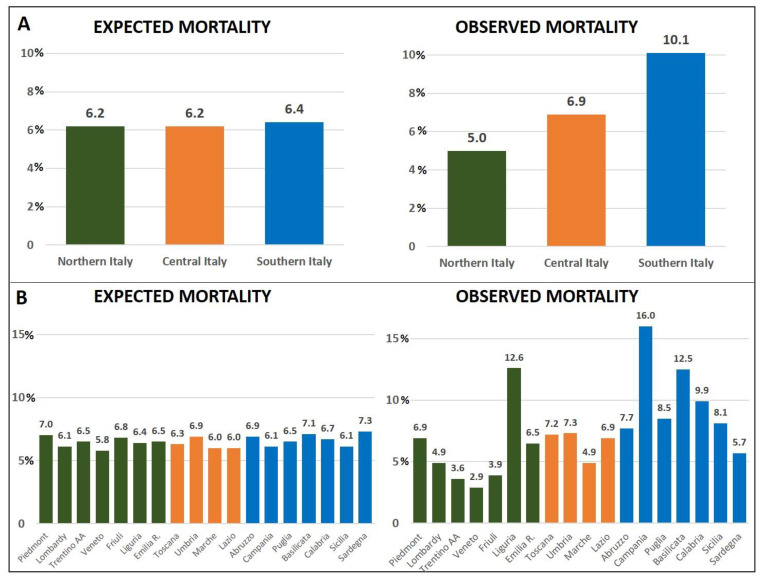
Expected and observed mortality rate according to different geographic areas (**A**) and regions (**B**).

**Figure 3 cancers-15-02429-f003:**
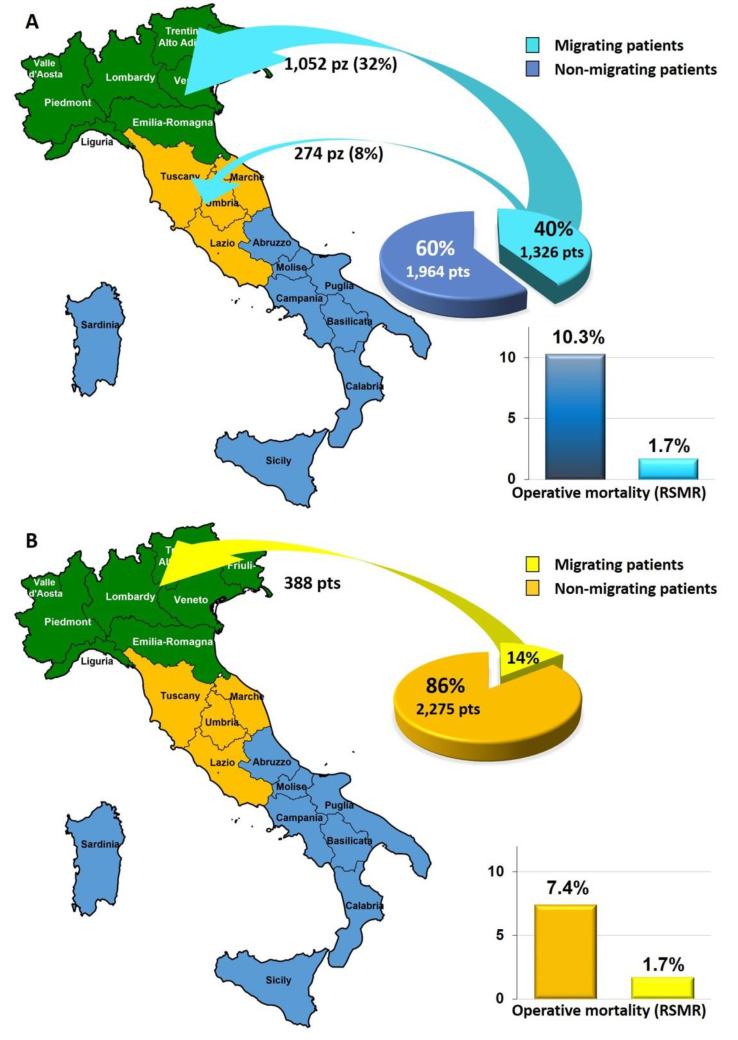
Impact of patients’ mobility from Southern (**A**) and Central Italy (**B**).

**Table 1 cancers-15-02429-t001:** Long-distance mobility of patients undergoing pancreatic resections from 2014 to 2016, according to the region/area of residence and the region/area where they received surgery. Short-distance mobility (inter-regional mobility within one hour’s travel time by car) was excluded. Escape rate and attraction rate of each region are reported.

Area/Region	Resident Population	Resident Patients Undergoing Resection		Resections in the Region/Area Out of Overall Resections in Italy		Migrating Patients * from a Specific Area/Region (Escape Rate ^a^)		Migrating Patients * to a Specific Area/Region (Attraction Rate ^b^)	
		N	‰	N	%	N	%	N	%
** *Northern Italy* **	27,774,970	6561	0.24	8023	63.4	75	1.1	1462	18.2
Piedmont	4,341,375	987	0.23	905	7.1	155	15.7	73	8.1
Valle d’Aosta	126,933	19	0.15	2	0.1	17	89.5	0	0
Lombardy	10,103,969	2441	0.24	3037	24.0	193	7.9	789	26.0
Trentino A.A.	1,074,819	210	0.20	165	1.3	53	25.2	8	4.8
Veneto	4,907,704	1089	0.22	2275	18.0	51	4.7	1237	54.4
Friuli V.G.	1,211,357	264	0.22	226	1.8	58	22.0	20	8.8
Liguria	1,543,127	417	0.27	261	2.1	173	41.5	17	6.5
Emilia-Romagna	4,467,118	1209	0.27	1152	9.1	214	17.7	157	13.6
** *Central Italy* **	11,986,958	2663	0.22	2638	20.8	388	14.6	363	13.8
Tuscany	3,722,729	1061	0.29	1096	8.7	118	4.1	153	14.0
Umbria	880,285	217	0.25	176	1.4	62	28.6	21	0.7
Marche	1,518,400	309	0.20	282	2.2	119	38.5	92	3.2
Lazio	5,865,544	1076	0.18	1084	8.6	174	16.2	182	16.8
** *Southern Italy* **	20,482,711	3290	0.16	2001	15.8	1326	40.3	37	1.8
Abruzzo	1,305,770	272	0.21	155	1.2	143	52.6	26	16.8
Molise	308,493	60	0.19	19	0.2	46	76.7	5	26.3
Campania	5,785,861	771	0.13	468	3.7	323	41.9	20	4.3
Puglia	4,008,296	669	0.17	481	3.8	240	35.9	52	10.8
Basilicata	556,934	84	0.15	56	0.4	35	41.7	7	12.5
Calabria	1,924,701	294	0.15	71	0.6	224	76.2	1	1.4
Sicily	4,968,410	838	0.17	559	4.4	294	35.1	15	2.7
Sardinia	1,630,474	302	0.19	192	1.5	112	37.1	2	1.0

* Migrating patients were operated on in a region different from the one where they reside and farther than one hour from the residence. ^a^ The escape rate was defined as the ratio between patients who migrated for surgery (from the region/area of residence) and overall patients who received resection residing in the same region/area. ^b^ The attraction rate was defined as the ratio between patients residing in another region/area and overall patients operated on in the region/area.

**Table 2 cancers-15-02429-t002:** General characteristics of migrating patients (undergoing long-distance mobility) and non-migrating patients (receiving surgery in the region where they reside).

	Migrating Patients	Non-Migrating Patients	*p*
**Number of patients**	2603 (20.6%)	10059 (79.4%)	
**Age > 70**	706 (27.0%)	4291 (42.7%)	<0.001
**Male gender**	1377 (52.9%)	5419 (53.9%)	0.376
**Charlson comorbidity index**			<0.001
0	2404 (92.4%)	8622 (85.7%)	
1	185 (7.1%)	1265 (12.6%)	
≥2	14 (0.5%)	172 (1.7%)	
**Type of resection**			0.041
Major (PD-TP)	1767 (67.9%)	7036 (69.9%)	
Minor (others)	836 (32.1%)	3023 (30.1%)	
**Hospital volume**			<0.001
Very low volume	141 (5.4%)	2232 (22.2%)	
Low volume	171 (6.6%)	2259 (22.5%)	
Medium volume	290 (11.1%)	2507 (24.9%)	
High volume	464 (17.8%)	1762 (17.5%)	
Very high volume	1537 (59.0%)	1299 (12.9%)	
**Private patients**			<0.001
No	2408 (92.5%)	9786 (97.3%)	
Yes	195 (7.5%)	273 (2.7%)	
**Diagnosis**			0.001
Pancreatic cancer and CP	1661 (63.8%)	6770 (67.3%)	
Other	942 (36.2%)	3289 (32.7%)	
**Laparoscopy**	203 (7.8%)	585 (8.5%)	0.230
**Multivisceral resection**	184 (7.1%)	1313 (13.1%)	<0.001
**Vascular resection**	140 (5.4%)	734 (7.3%)	0.001

Migrating patients were operated on in a region different from the region where they reside and farther than one hour from the residence. PD: pancreaticoduodenectomy; TP: total pancreatectomy; CP: chronic pancreatitis.

**Table 3 cancers-15-02429-t003:** Univariate and multivariate analyses of factors associated with operative mortality.

		Univariate Analysis			Multivariate Analysis		
Variable	In-Hospital Mortality	OR	95%CI	*p*-Value	OR	95%CI	*p*-Value
**Age (years)**							
≤70	341 (4.4%)	1			1		
>70	448 (9.0%)	2.115	1.83–2.45	**<0.001**	1.923	1.66–2.23	**<0.001**
**Gender**							
Male	471 (6.9%)	1			1		
Female	318 (5.4%)	0.770	0.66–089	**<0.001**	0.784	0.87–0.91	0.001
**Charlson comorbidity index**							
0	646 (5.9%)	1			1		
1	103 (7.1%)	1.229	0.99–1.52	0.061	0.993	0.80–1.24	0.953
≥2	40 (21.5%)	4.402	3.07–6.30	**<0.001**	3.680	2.53–5.35	<0.001
**Hospital Volume**							
Very low volume	252 (10.6%)	1			1		
Low volume	170 (7.0%)	0.633	0.52–0.78	**<0.001**	0.632	0.51–0.78	<0.001
Medium volume	170 (6.1%)	0.545	0.44–0.67	**<0.001**	0.568	0.46–0.70	<0.001
High volume	110 (4.9%)	0.438	0.35–0.55	**<0.001**	0.516	0.40–0.66	<0.001
Very high volume	87 (3.1%)	0.266	0.21–0.34	**<0.001**	0.399	0.30–0.53	<0.001
**Geographical area (operating center)**							
Northern Italy	404 (5.0%)	1			1		
Central Italy	182 (6.9%)	1.398	1.17–1.67	<0.001	1.323	1.10–1.59	0.003
Southern Italy	203 (10.1%)	2.129	1.78–2.54	<0.001	1.625	1.34–1.97	<0.001
**Geographical area (patient’s residence)**							
Northern Italy	370 (5.6%)	1					
Central Italy	177 (6.6%)	1.206	1.01–1.45	0.048			
Southern Italy	239 (7.3%)	1.327	0.12–1.57	0.001			
**Patient mobility**							
Non-migrating patients	709 (7.0%)	1			1		
Migrating patients	80 (3.1%)	0.418	0.33–0.53	**<0.001**	0.698	0.54–0.90	**0.006**
**Type of resection**							
Minor (others)	141 (3.7%)	1			1		
Major (PD-TP)	648 (7.4%)	2.095	1.74–2.52	**<0.001**	1.770	1.46–2.14	<0.001
**Diagnosis**							
Pancreatic cancer and CP	522 (6.2%)	1					
Other	267 (6.3%)	1.021	0.88–1.19	0.794			
**Laparoscopy**							
No	781 (6.7%)	1			1		
Yes	8 (0.8%)	0.105	0.05–0.21	**<0.001**	0.150	0.07–0.30	<0.001
**Private patient**							
No	779 (6.4%)	1			1		
Yes	10 (2.1%)	0.320	0.17–0.60	<0.001	0.385	0.20–0.73	**0.003**
**Vascular resection**							
No	742 (6.3%)	1					
Yes	47 (5.4%)	0.846	0.62–1.15	**0.280**			
**Multivisceral resection**							
No	680 (6.1%)	1					
Yes	109 (7.3%)	1.211	0.98–1.49	0.074			

OR: odds ratio; PD: pancreaticoduodenectomy; TP: total pancreatectomy; CP: chronic pancreatitis.

**Table 4 cancers-15-02429-t004:** Mortality (observed mortality and risk-standardized mortality rate—RSMR) of pancreatic resections per region and geographic area, with stratification between migrating and non-migrating patients.

	Mortality of Resections Performed in Each Area/Region				Mortality According to the Area/Region of Patients’ Residence								Mortality of Migrating Patients According to the Area/Region Where They Received Surgery			
					Migrating Patients *				Non-Migrating Patients °							
Area/Region	No. of Resections	Observed Mortality		RSMR	No. of Resections	Observed Mortality		RSMR	No. of Resections	Observed Mortality		RSMR	No. of Resections	Observed Mortality		RSMR
		No.	%	%		No.	%	%		No.	%	%		No.	%	%
** *Northern Italy* **	8023	404	5.0	5.1	75	2	2.7	1.4	6561	368	5.6	6.0	1462	36	2.5	1.3
Piedmont	905	62	6.8	6.1	100	1	3.0	1.6	887	59	6.7	6.7	61	5	8.2	3.4
Lombardy	3037	151	5.0	5.1	108	2	1.9	1.0	2333	124	5.3	6.0	725	22	3.0	1.7
Trentino AA	165	6	3.6	3.5	34	3	8.8	5.3	175	5	2.9	3.2	7	2	28.5	13.3
Veneto	2275	68	3.0	3.2	38	2	5.2	3.1	1051	51	4.9	5.5	1121	17	1.5	0.9
Friuli	226	9	4.0	3.7	56	0	0	0	208	8	3.9	3.9	9	0	0	0
Liguria	261	33	12.6	12.2	165	8	4.8	2.3	252	33	13.1	14.4	11	1	9.1	3.9
Emilia-Romagna	1152	75	6.5	6.3	184	3	1.6	0.9	1025	69	6.7	7.2	140	6	4.3	2.3
** *Central Italy* **	2638	182	6.9	7.0	388	12	3.1	1.7	2275	165	7.3	7.9	363	17	4.7	2.5
Tuscany	1096	80	7.3	7.3	110	3	2.7	1.5	951	75	7.9	8.7	144	7	4.5	2.8
Umbria	176	13	7.4	6.7	58	1	1.7	1.0	159	12	7.5	7.7	11	1	9.1	5.6
Marche	282	14	5.0	5.1	112	6	5.4	3.1	197	14	7.1	8.0	89	0	0	0
Lazio	1084	75	6.9	7.2	162	3	1.9	1.0	914	63	6.9	8.1	174	11	6.3	3.4
** *Southern Italy* **	2001	203	10.1	9.9	1326	38	2.9	1.5	1964	201	10.2	10.8	37	2	5.4	2.6
Abruzzo	155	12	7.7	7.0	134	8	6.0	3.4	138	12	8.7	9.0	20	0	0	0
Campania	468	75	16.0	16.4	320	8	2.5	1.4	451	74	16.4	18.8	20	3	15	8.7
Puglia	481	41	8.5	8.2	239	5	2.1	1.3	430	37	8.6	9.3	44	4	9.1	4.8
Basilicata	56	7	12.5	11.0	27	0	0	0	57	6	10.5	10.5	4	0	0	0
Calabria	71	7	9.9	9.1	224	10	4.5	2.4	70	7	10	10.4	1	0	0	0
Sicily	559	45	8.1	8.2	294	8	2.7	1.5	544	44	8.1	9.4	15	1	6.7	2.9
Sardinia	192	11	5.7	4.9	112	1	0.9	0.5	190	11	5.8	5.6	2	0	0	0

* Migrating patients were operated on in a region different from the one where they reside and farther than one hour from the residence. ° Non-migrating patients received surgery in the region where they reside. RSMR: risk-standardized mortality rate. NB: Overall mortality rate = 6.23, overall mortality rate (non-migrating) = 7.1, overall mortality rate (migrating) = 3.1.

**Table 5 cancers-15-02429-t005:** Availability of hospitals performing pancreatic resections per area/region.

Area/Region	Overall Hospitals Performing Pancreatic Surgery	Minimum Volume ≥ 10 Resections/Year	Minimum Volume ≥ 10 Resections/Year and Mortality ≤ 10%		Minimum Volume ≥ 10 Resections/Year and Mortality ≤ 5%	
			No.	Rate Per Million Inhabitants	No.	Rate Per Million Inhabitants
** *Northern Italy* **	**185**	55	46	1.65	32	1.15
Piedmont	27	5	4	0.92	3	0.69
Valle d’Aosta	1	0	0	0	0	0
Lombardy	79	23	17	1.68	13	1.28
Trentino AA	4	3	3	2.8	2	1.87
Veneto	27	8	8	1.63	7	1.42
Friuli	18	3	3	2.48	2	1.65
Liguria	9	2	1	0.64	0	0
Emilia-Romagna	20	11	10	2.24	5	1.12
** *Central Italy* **	85	17	15	1.25	5	0.42
Tuscany	28	5	4	1.07	0	0
Umbria	9	2	2	2.27	0	0
Marche	9	2	2	1.31	1	0.66
Lazio	39	8	7	1.19	4	0.68
** *Southern Italy* **	126	20	14	0.68	7	0.34
Abruzzo	12	1	1	0.77	1	0.77
Molise	3	0	0	0	0	0
Campania	32	5	1	0.17	0	0
Puglia	27	4	3	0.75	1	0.25
Basilicata	3	1	0	0	0	0
Calabria	10	1	1	0.52	0	0
Sicily	31	6	6	1.21	4	0.80
Sardinia	8	2	2	1.23	1	0.61

## Data Availability

Data are available from the corresponding author on reasonable request.

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
