# Peer review of "Geographical Disparities and Patients’ Mobility: A Plea for Regionalization of Pancreatic Surgery in Italy"

_cancers, 2023, doi:10.3390/cancers15092429_

Round 1

Reviewer 1 Report

This manuscript is very well-written and interesting. I congratulate the authors for this excellent study.  The impact of regionality on medical outcomes is the main question addressed by the research. The study is original and  very interesting. . Additionally, the same issue is discussed in my country. The remote surgery may resolve but further study is needed. I have never seen the similar article and the study would help the similar study in other countries. The more detailed data such as patient characteristics and the tumor status should be considered. But it will be difficult to achieve such data in nation wide survey.  The conclusion is based on the results and may suggest some answers to the main question. The references are appropriate.

The figure legends of Fig. 3 is recommended to presented clearly as follows; Figure 3. impact of patients’ mobility from Southern (A) and Central Italy (B).

Author Response

Reviewer 1

This manuscript is very well-written and interesting. I congratulate the authors for this excellent study.  The impact of regionality on medical outcomes is the main question addressed by the research. The study is original and  very interesting. . Additionally, the same issue is discussed in my country. The remote surgery may resolve but further study is needed. I have never seen the similar article and the study would help the similar study in other countries. The more detailed data such as patient characteristics and the tumor status should be considered. But it will be difficult to achieve such data in nation wide survey.  The conclusion is based on the results and may suggest some answers to the main question. The references are appropriate.

The figure legends of Fig. 3 is recommended to presented clearly as follows; Figure 3. impact of patients’ mobility from Southern (A) and Central Italy (B).

We thank referee 1 for the favorable judgement. The legend of figure 3 has been changed as suggested.

Reviewer 2 Report

I read with great interest the manuscript submitted by Balzano et al. The authors gave a comprehensive overview of volume distribution of pancreatic surgery in Italy, focusing in particular on patients’ mobility and its effect on post-operative mortality. According to the data presented, there is an urgent need of implementing facilities for pancreatic surgery in southern and central regions. Although the topic is of great interest, there are some issues that need to be solved before acceptance:
1. the multivariate analysis (Table3) need to be revised. Univariate analysis was significant for geographical area distribution (p=0.048; p=0.001). However, this feature was not included in the multivariate analysis. Similarly, Charlson comorbidity index was not statistically significant at the univariate analysis but put, anyway, into the multivariate model. Is there any justification? I would recommend to reperform the analysis;
2. English should be extensively revised. In particular, some Italian word is still present in the manuscript (i.e. table 3: “struttura” and “residenza”);
3. In the discussion section, the authors mention the economic implication due to the patients’ mobility (lines 336-341). What does that mean? if costs vary according to the region, it would be interesting to estimate costs in relation to the migration.
4. Since the authors proposed to classify surgical centers according to the caseload in different volume categories I would suggest to report the distribution of the hospitals in Italy according to such classification. How many very high and high volume centers in north center and south? How the volume impacted on mortality? Is mortality rate comparable between centers according to the volume caseload independently of the geographic location? I think this is an essential information in order to avoid a generalization in the conclusions (north is better than center and south regions)

Author Response

Reviewer 2

I read with great interest the manuscript submitted by Balzano et al. The authors gave a comprehensive overview of volume distribution of pancreatic surgery in Italy, focusing in particular on patients’ mobility and its effect on post-operative mortality. According to the data presented, there is an urgent need of implementing facilities for pancreatic surgery in southern and central regions.

We thank the reviewer for the positive comment

Although the topic is of great interest, there are some issues that need to be solved before acceptance:

  1. the multivariate analysis (Table3) need to be revised. Univariate analysis was significant for geographical area distribution (p=0.048; p=0.001). However, this feature was not included in the multivariate analysis. Similarly, Charlson comorbidity index was not statistically significant at the univariate analysis but put, anyway, into the multivariate model. Is there any justification? I would recommend to reperform the analysis;

We thank the reviewer for highlighting this point. The statistical model reported in table 3 was conducted using binary logistic regression. The multivariate model was obtained by a stepwise backward conditional elimination (Wald method accounting a p> 0.010 for elimination). In this way, all the variable reported in the univariate analysis were entered in the model. The multivariate analysis reported only the variable retained at the last step of the multivariate model. Considering the wide sample size reported, backward elimination seemed the most appropriate method for conducting the analysis. Given the high collinearity between the geographical area of the operating center and the patient’s geographical area of residency it is not surprising that the residency has been excluded by the stepwise elimination.

The following paragraph to clarify the statistical methodology was added to the methods:

“Multivariate analysis for factors influencing in-hospital mortality was conducted using logistic regression. The model was obtained through a stepwise backward elimination entering all the variables analyzed at univariate (Wald method with p> 0.010 for removal from the model)”

  1. English should be extensively revised. In particular, some Italian word is still present in the manuscript (i.e. table 3: “struttura” and “residenza”)

English text was revised, and Italian words have been changed.

  1. In the discussion section, the authors mention the economic implication due to the patients’ mobility (lines 336-341). What does that mean? if costs vary according to the region, it would be interesting to estimate costs in relation to the migration.

The reviewer focused another crucial topic related to centralization of complex care. Low socioeconomic classes cannot afford the expenses of migration, so the costs of mobility for cure are an ethical topic. In Italy, there are two kinds of economic implications due to patients’ mobility.

The first one happens in every country: it is the direct out-of-pocket expenses of patients and relatives to receive care at long distance from home (travel expenses, costs for accompanying relatives and caregivers staying away from home). We do not have the ability to retrospectively estimate expenses incurred by the patient and family. The present study derived from administrative data recorded by the Italian Minister of Health. Only region of patients’ residency could be poled from these data. No information is retrievable about the cost of migration. We can only speculate on this point, but many variables should be accounted such as: the type of transport the patients and his relatives use to reach the operating center, how many trips the patient must perform before surgery, a possible lodgment for the family near the operating center, the loss of profit due to work discontinuation. Given this complexity, it is impossible for us to estimate the costs of patients’ out-of-pocket expenses.

The second economic issue we refer to in the manuscript concerns the transfer of financial resources from poorer to richer regions. This is a specific problem of Italy, due to the organization of the Italian National Health Service. Patient mobility to receive care in other regions is allowed by the Italian NHS (without costs for the patient), but each region has to pay for the treatment provided to residents by facilities located in other regions. The reimbursement for each operation is similar in all Italian regions, but patients’ mobility causes a transfer of money from poorer to richer regions, to pay for operations received from their residents in other regions. We believe that the quantification of this transfer of money is beyond the scope of this study.

  1. Since the authors proposed to classify surgical centers according to the caseload in different volume categories I would suggest to report the distribution of the hospitals in Italy according to such classification. How many very high and high volume centers in north center and south?

This is an important point highlighted by the reviewer. In a previous article we included one figure showing the geographical distribution of Italian hospitals with minimal volume of 10 resections/year and mortality below 5% or 10%  (Balzano G et al. Modelling centralization of pancreatic surgery in a nationwide analysis. Br J Surg. 2020 Oct;107(11):1510-1519. doi: 10.1002/bjs.11716. PMID: 32592514.). We have not reproduced such figure in the present article, but we have calculated the availability of hospitals performing pancreatic resections with adequate outcome per area/region (table 5). This table summarizes the information requested by the reviewer.

How the volume impacted on mortality?

The impact of volume on mortality of pancreatic surgery in Italy was detailed in previous articles (1. Balzano G et al. Modelling centralization of pancreatic surgery in a nationwide analysis. Br J Surg. 2020 Oct;107(11):1510-1519. doi: 10.1002/bjs.11716. PMID: 32592514; 2. Balzano G et al. Overuse of surgery in patients with pancreatic cancer. A nationwide analysis in Italy. HPB (Oxford). 2016 May;18(5):470-8. doi: 10.1016/j.hpb.2015.11.005. Epub 2016 Feb 5. PMID: 27154812; PMCID: PMC4857063; 3. Balzano G et al. Effect of hospital volume on outcome of pancreaticoduodenectomy in Italy. Br J Surg. 2008 Mar;95(3):357-62. doi: 10.1002/bjs.5982. PMID: 17933001.). Also in the present analysis (Table 3), hospital volume confirmed an independent effect on mortality.

Is mortality rate comparable between centers according to the volume caseload independently of the geographic location? I think this is an essential information in order to avoid a generalization in the conclusions (north is better than center and south regions)

This is another interesting comment. In the manuscript we indicated that the main problem causing differences among geographical areas is the lack of high-volume hospitals in southern regions, and (to a lesser extent) in central Italy. In table 5, the number of overall hospitals performing >10 resections/year per region and per geographic area is reported, as well as the number of hospitals combining such volume with low mortality. 

Reviewer 3 Report

The authors analyze a topic which is of interest in most countries.

The presentation is clear, comprehensive and well documented.

The references are appropriate, up-to-date and contain 32 titles.

The self-citations reflect the interest of the author ( dr Balzano) on the topic without overlapping.

The figures (3) are appropriate and mandatory for sustaining the topic .

The 5 tables offer concentrated information on the topic. Tabel 3 contains Italian words -Geographical area (struttura) and Geographical area (residenza). Tabel 4 has to be properly placed in the page.

I found no plagiarism.

The discussions and conclusions are coherent and connected to the content.

In my opinion the paper fits the journal and the language is correct and understandable.

I recommend the paper to be accepted.

Author Response

Reviewer 3

The authors analyze a topic which is of interest in most countries.

The presentation is clear, comprehensive and well documented.

The references are appropriate, up-to-date and contain 32 titles.

The self-citations reflect the interest of the author ( dr Balzano) on the topic without overlapping.

The figures (3) are appropriate and mandatory for sustaining the topic .

The 5 tables offer concentrated information on the topic. Tabel 3 contains Italian words -Geographical area (struttura) and Geographical area (residenza). Tabel 4 has to be properly placed in the page.

I found no plagiarism.

The discussions and conclusions are coherent and connected to the content.

In my opinion the paper fits the journal and the language is correct and understandable.

I recommend the paper to be accepted.

We thank referee 3  for the positive review. We removed Italian words from table 3